# Influence of Dopant Concentration and Annealing on Binary and Ternary Polymer Blends for Active Materials in OLEDs

**DOI:** 10.3390/nano12224099

**Published:** 2022-11-21

**Authors:** Maria Gioti, Despoina Tselekidou, Vasileios Foris, Vasileios Kyriazopoulos, Kyparisis Papadopoulos, Spyros Kassavetis, Stergios Logothetidis

**Affiliations:** 1Nanotechnology Lab LTFN, Department of Physics, Aristotle University of Thessaloniki, GR-54124 Thessaloniki, Greece; 2Organic Electronic Technologies P.C. (OET), 20th KM Thessaloniki—Tagarades, GR-57001 Thermi, Greece

**Keywords:** white OLED, blend, binary, ternary, Förster

## Abstract

Obtaining white light from organic LEDs is a considerable challenge and, to realize white light emission, many studies have been conducted, primarily addressing two- or three-color blend systems as a promising strategy. In this work, pristine films, grown by spin coating, consisting of commercial blue Poly(9,9-di-n-octylfluorenyl-2,7-diyl) (PFO), green Poly(9,9-dioctylfluorene-alt-benzothiadiazole) (F8BT), and red spiro-copolymer (SPR) light-emitting materials, were studied as reference materials. Afterward, binary (SPR doped in host PFO) and ternary (SPR and F8BT doped in host PFO) thin films were successfully prepared with various ratios. The characterization of the as-grown and thermally-treated blend films was focused on their optical and photophysical properties. After, the fabrication of OLED devices on glass substrates was carried out for the evaluation of a blend’s composition and annealing in terms of the devices’ electrical characteristics and electro-emission properties in order to achieve white light emission. Their analysis provided insights into the energy transfer mechanisms between the constituent materials, which were correlated to host–guest interactions as well as to the structural changes originated by thermal treatment, leading to the crystallization of PFO. Finally, the OLEDs based on ternary blends approach the white light emission with (x, y) of (0.272, 0.346). These fabricated devices also exhibit turn-on voltages as low as 3 V, accompanied by remarkable luminance values above 3000 cd/m^2^.

## 1. Introduction

Organic light emitting diodes (OLEDs), with emphasis on white OLEDs (WOLEDs), have many attractive features, such as: they can be easily fabricated using wet processes on flexible substrates, they are cost-effective, and they perform well in many applications, from flat panel displays to interior lighting [1,2,3,4,5]. Solution-processed WOLEDs are superior to traditional light sources in terms of their merits in ensuring pure white-light emission, low-energy consumption, large area, low-cost thin-film fabrication, etc. To produce WOLEDs, at least two or three emissive materials are deposited in a multilayer structure [6,7], or, alternatively, single-layer emitting copolymers that bear different emitting chromophores [8,9,10,11] or single-layer mixed polymers by blending (or doping) [2,12,13,14,15] can be implemented. Unfortunately, the fabrication of WOLEDs based on multilayered emissive layers involves the difficulty of finding a suitable solvent for each layer without dissolving the underlying layers, and the produced devices can suffer from some remarkable disadvantages, such as intricate device fabrication and voltage-dependent emission color, etc. Therefore, it is preferable to fabricate WOLEDs with a simpler architecture, that is, the use of single-layer emitting materials. However, the achievement of highly efficient WOLEDs by using single-layer white emitters is still a challenge in terms of both the judicious design of the light emitters and their device architectures. Thus, much work should still be done to develop new types of white emitters as well as to fabricate highly efficient WOLEDs by combining the merits of a simple fabrication process, low-energy consumption, and low cost.

Förster resonance energy transfer (FRET) excitation in a blend requires the appropriate mixing of the two or three materials as well as the optimum spectral overlap between acceptor absorption and donor emission. By successful mixing of donors and acceptors, in blended emitting materials, the concentration quenching of the produced excitons is reduced, enabling tuning of the Commission Internationale de l’Enclairage (CIE) coordinates of light emission and enhancing efficiency [16,17,18,19]. 

Theoretically, white light emission can be achieved using the combination of two different emitting colors, either via a combination of blue and red emitting colors or through the combination of three colors, namely red, green, and blue [20,21,22]. The latter offers a better way of producing white light for OLEDs. Al-Asbahi [13] suggested the optimum combination of poly (9,9′-di-n-octylfluorenyl-2,7-diyl) (PFO) blue emitter, 2-butyl-6-(butylamino) benzo (de) isoquinoline-1,3-dione (F7GA) green emitter, and poly (2methoxy-5-(2-ethylhexyloxy)-1,4-phenylenevinylene) (MEH-PPV) red emitter in ternary blended thin films to obtain white OLEDs, exhibiting a maximum luminance of 2295 cd/m^2^ and CIE coordinates of (0.31, 0.24). However, there is still space for further optimization in terms of white color coordinates and devices’ operational characteristics. Recently, we reported the fabrication of WOLED devices based on blends of PFO, poly (9,9-dioctylfluorene-altbaenzothiadiazole) (F8BT) green emitter, and poly (2-methoxy-5-(30,70-dimethylocty loxy)-1,4-phenylenevinylene) (MDMO-PPV) red emitter exhibiting high color stability but relatively low brightness [23].

This paper addresses the use of an alternative combination of emitting materials to produce either binary or ternary blends for the manufacturing of WOLED devices with CIE coordinates as close as possible to the ideal ones (0.33, 0.33), together with higher efficiency. These materials are the PFO as the blue emitter and a spiro-copolymer (SPR) of high efficiency as the red one, for the case of binary blends. Additionally, F8BT was used, for the case of ternary blends, as the green emitter. The SPR red emitter used in the present work has an absorption profile quite different from that usually used in blends’ poly(phenylenevinylene) (PPV) derivatives, such as MEH-PPV or MDMO-PVV. A detailed study was conducted to provide insights into the energy transfer mechanism between the PFO donor and the SPR acceptor, or the PFO donor and the F8BT and SPR acceptors. The effect of annealing on the pristine films and the produced binary and ternary blends was also investigated, as it has been shown that thermal annealing can remarkably affect the device’s performance.

## 2. Materials and Methods

The poly (9,9-di-n-octylfluorenyl-2,7-diyl) PFO (Mw = 114.050) and the poly (9,9-dioctylfluorene-alt-benzothiadiazole) F8BT (Mw = 237,460) were purchased from Ossila (Sheffield, UK). The red light-emitting spiro-copolymer-001 SPR (Mw = 180,000) was purchased from Sigma-Aldrich Chemie GmbH, Taufkirchen, Germany. All materials were dissolved in a toluene solvent. For the Hole Transport Layer (HTL), a solution of poly-3,4-ethylene dioxythiophene:poly-styrene sulfonate (PEDOT:PSS, Clevios Heraus, Hanau, Germany) AI 4083 mixed with ethanol in the ratio of 2:1 was prepared. Glass and pre-patterned Indium-Tin Oxide-coated (ITO) glass substrates (received by Ossila, Sheffield, United Kingdom) were extensively cleaned by sonication in DI, acetone, and ethanol for 10 min followed by drying with nitrogen. The substrates were also treated with oxygen plasma at 40 W for 3 min. Glass substrates were used to deposit all samples for measuring absorption by Spectroscopic Ellipsometry (SE), Absorption Spectroscopy (AS), and Photoluminescence (PL), whereas the ITO/glass substrates were used for the fabrication of OLED devices for Electroluminescence (EL) measurements.

SPR and PFO solutions with different weight ratios, stirred under heating for 24 h, were used to prepare the homogeneous binary blends. In the same manner, SPR, F8BT, and PFO solutions were used to prepare the ternary blends. The precise ratios used are listed in Table 1. For all solutions, the final concentration was fixed at 1% *w*/*w*.

Both binary and ternary blends were deposited onto the glass and etched ITO substrates (2 cm × 1.5 cm) by the spin-coating technique at a rotational speed of 1500 rpm for 1 min. Two separate samples of each of the two pristine and blended materials were developed to proceed to the comparative characterization of as-grown and thermally treated films. The annealed films were baked in a vacuum oven for 10 min at 120 °C. The thickness of all films was calculated by SE data analysis. The thicknesses of the pristine films and of representative blended films are listed in Table 2.

For the device fabrication, the PEDOT:PSS HTL layer was deposited by the spin coating method onto the glass/ITO substrate followed by annealing at 120 °C for 5 min. The emitting layers (EML) were spun using the binary and ternary blend solutions onto the PEDOT:PSS layer. Finally, a bilayer of 6 nm thick Ca and 125 nm thick Ag was used as a cathode electrode bilayer and was deposited using the appropriate shadow masks by Vacuum Thermal Evaporation (VTE). In Figure 1a,b, the structures of the deposited monolithic films and of the fabricated multilayer OLED devices are illustrated, respectively. The highest occupied molecular orbital (HOMO) and lowest unoccupied molecular orbital (LUMO) level for each polymer studied in this work and used for the blends’ formation, as well as the energy levels of the sequential layers, namely ITO—anode, PEDOT:PSS—HTL, and Ca/Ag—cathode bilayer, of the OLED devices, are presented in Figure 1c.

The blend films and the OLED devices were characterized in terms of their optical, photophysical, and electrooptical properties using Spectroscopic Ellipsometry (SE) and Absorption, Photoluminescence, and Electroluminescence Spectroscopies.

The SE measurements were conducted using a phase modulated ellipsometer (Horiba JobinYvon, UVISEL, Europe Research Center—Palaiseau, France) from the near IR to far UV spectral region 1.5–6.5 eV with a step of 20 meV at a 70° angle of incidence. The SE experimental data were fitted to model-generated data using the Levenberg–Marquardt algorithm, taking into consideration all the fitting parameters of the applied model.

The absorbance measurements were conducted using the set-up of ThetaMetrisis (model FR UV/VIS) (ThetaMetrisis S.A., Athens, Greec). Combining a deuterium and halogen light source, all measurements were performed in the 300–700 nm spectral range.

The Photoluminescence and Electroluminescence measurements of the active layers and the final OLED devices, respectively, were measured using the Hamamatsu Absolute PL Quantum Yield measurement system (C9920-02) and the External Quantum Efficiency (EQE) system (C9920-12) (Joko-cho, Higashi-ku, Hamamatsu City, 431-3196, Japan), which measures the luminance and light distribution of the devices.

## 3. Results and Discussion

### 3.1. Optical Properties

SE is a powerful, robust, non-destructive, and surface-sensitive optical technique for the determination of the optical properties as well as the thickness of the thin films, in our case, of the light-emitting polymers and blends. Through the SE technique, we can measure the pseudodielectric function:〈*ε*(*E*)〉 = 〈*ε*_1_(*E*)〉 + *i*〈*ε*_2_(*E*)〉(1)
of the studied thin films. By applying the appropriate modelling and fitting procedures, we can extract significant information about the dielectric function ε(E), the thickness of the thin films with nanometer-scale precision, the refractive index, the absorption coefficient, and the optical constants, such as the fundamental band gap and the higher energy optical gaps.

The dielectric functions of PFO, F8BT, and SPR and of their blends were calculated through the analysis of the measured 〈ε(E)〉 spectra, using the modified Tauc–Lorentz (TL) dispersion oscillator model, which has been successfully applied in amorphous organic semiconductors [8,24,25,26,27] in combination to the three-phase geometrical model glass/emitting film/air (see Figure 1a). Particularly, in the case of organic materials, electron–phonon coupling is of great importance [24] and, thus, for the best description of the optical response of amorphous organic thin films, the energy-dependent broadening is employed in the TL oscillator model. The model and SE analysis procedure are presented in detail in Ref. [28].

The real ε_1_(E) and the imaginary ε_2_(E) parts of the calculated complex ε(E), using the best-fit parameters, of the pristine PFO, F8BT, and SPR films and the representative Binary#1 and Ternary#1 blend films are plotted versus photon energy in Figure 2a,b, respectively. It is evident that the characteristic absorptions of the blends are similar to that of pure PFO because of the low contents of F8BT and SPR. Figure 2c,d depicts the dielectric functions of the as-grown and annealed blends, Binary#1 and Ternary#1. The thermally treated films, annealed at 120 °C, exhibit a shift in the absorption edge to lower energies and an increase in the strength of the characteristic absorption features. These modifications are considered to the presence of small contents of the crystalline PFO phase in the amorphous PFO matrix after thermal annealing, which has an extended conjugation length [29].

The complex dielectric function is directly related to complex refractive index N:(2)$$\varepsilon \left(E\right)=N{\left(E\right)}^{2}$$
that includes two components, n and κ. The n component represents a quantitative description of light propagation in the medium, and the κ component stands for the extinction coefficient, which expresses absorption of light by medium [26]:(3)$$N\left(E\right)=n+i\kappa$$

The absorption coefficient is mathematically expressed as follows [26]:(4)$$\alpha =\frac{4\pi \kappa }{\lambda }$$
where λ is the wavelength.

More often, absorbance A of thin films is characterized by UV–vis-NIR AS. The absorbance of a thin layer is proportional to its thickness and to the absorption coefficient α of the material:(5)A=α d 

Absorption spectroscopy is an analytical method commonly used for studying interactions of light with materials, which can be interpreted by changes in absorption spectra of the materials.

The shapes of the determined absorption coefficients spectra of the SPR, F8BT, and PFO were compared with the shapes of the absorption spectra obtained by UV–vis AS. Figure 3 shows the normalized absorbance spectra and the absorption coefficient spectra. The latter were calculated through the analysis of the SE data. There are some differences in the shapes of the spectra, probably due to the uncertainties of experimental instruments and to layers’ inhomogeneity. The characteristic features in each material appear in the corresponding spectra with a slight blue-shift of the absorption coefficient compared to the absorbance. In addition, the absorption edge for all the films appears to be more abrupt in the case of the absorption coefficient. PFO demonstrated the first absorption maximum within the range of 360–380 nm, while F8BT was within the range of 410–450 nm. SPR showed a characteristic plateau in absorption, which covers the range 310–460 nm.

The calculated absorption coefficient, derived through SE, which is regarded as the most accurate method for the precise determination of the optical properties of thin films, can also give us a quantitative comparison in terms of the strength of the absorption and not only compare the characteristic wavelength regions where the characteristic absorption peaks appear, such as absorbance. Therefore, it is more preferable to use the absorption coefficient in the study of spectral overlaps between absorption and photoemission of materials.

On the other hand, we have to notice that the band gap ($E_{g}$) derived by the SE analysis is not adequately estimated since, in the TL model, the discrete absorption features below the bandgap are not considered. However, the defects, or the polymer’s structural changes and disorder, contribute to the absorption tail at the absorption edge region [30,31]. Thus, for the precise estimation of the band gap, the Tauc plot analysis is applied, which assumes that the energy-dependent absorption coefficient *α* can be expressed for the case of the direct transition band gap, by the following equation [31]:(6)$${\left(a\cdot E\right)}^{2}=B\left(E-E_{g}^{Tauc}\right)$$
where E is the photon’s energy, $E_{g}^{Tauc}$ is the band gap energy, and B is a constant. In the resulting plot, the distinct linear regime denotes the onset of absorption, which is linearly fitted, and the energy of the optical band gap of the material is obtained.

Table 2 summarizes the results for the calculated band gaps $E_{g}$ and $E_{g}^{Tauc}$ and their energy difference (Δ*E*) of the pristine materials and the representative Binary#1 and Ternary#1 blends. It is evident from these results that the absorption edge is almost sharp, and the defects and disorder should be relatively low. Therefore, they do not have a strong contribution to the calculation of the absorption coefficient, which is not the case when measuring the absorbance. Furthermore, lower band gap $E_{g}$ values are systematically observed compared to the corresponding $E_{g}^{Tauc}$ values, either for the as-grown or thermally treated blends.

When different organic compounds, such as polymers, form a blended system, this system may undergo alteration in photophysical properties, including the energy transfer mechanism. In the photoluminescence process, radiative and non-radiative energy transfer may take place. In the radiative energy transfer mechanism, the energy is transferred from one molecule to another molecule in the form of electromagnetic radiation [32,33,34]. On the other hand, in the non-radiative energy transfer mechanism, the electronic interaction between two different organic materials takes place, that is, the excited energy of the electron from one molecule (donor) is transferred to the other molecule (acceptor) without emitting a photon, resulting in the quenched emission of the donor. The prerequisite for the non-radiative energy transfer mechanism to occur is the required overlay of the emission wave functions of the donor with that of the absorption of the acceptor. This overlap is not a unique condition for the occurrence of non-radiative transfer of the Förster type, which still depends on the distance between chromophores [35,36,37,38] that cannot be larger than the Förster radius and spatial orientation between electric dipoles of the involved electronic states [39,40,41,42].

### 3.2. Correlation of Absorption and Photoluminescence for Förster Resonance Energy Transfer (FRET)

The non-radiative energy transfer mechanism between PFO and F8BT is evident from the significant spectral overlap between the PFO donor emission and the F8BT acceptor absorption (Figure 4a). Here, it is noteworthy to mention that this huge spectral overlap between the emission of PFO and the absorption of F8BT manifests that a large fraction of energy shall be transferred to the latter. A respective sufficient spectral overlap, as shown in Figure 4b, also suggests the possible non-radiative energy transfer between PFO and SPR. On the contrary, no significant energy transfer is expected between F8BT and SPR, as demonstrated by Figure 4c. Figure 4d summarizes the overlaps between all three components used for the ternary systems. One can expect a cascade energy transfer from PFO to F8BT and then SPR. Finally, Figure 4e shows the correlation between the Absorbance and PL emission of the as-grown and thermally treated PFO films, while the inset shows the respective absorption coefficients. The observed red-shift in both the absorption edge and PL emission is consistent with the formation of crystalline phases upon annealing. More specifically, the PL spectrum of the as-grown PFO thin film has a main broad peak, which is deconvoluted with two individual peaks at 427 and 444 nm, as well as two additional weaker peaks centered at 469 and 514 nm, leading to a green emitting tail. For the case of the annealed PFO thin film, the corresponding PL peaks become significantly more distinct and red-shift to 434 and 457 nm. It is well established by the literature that thermal annealing has a strong influence on the fluorescence characteristics of the pure PFO film, which is attributed to the formation of crystalline phases of the polymer within the layer [43,44].

### 3.3. Photoluminescence of the Binary and Ternary Thin Films

Figure 5 shows the photoluminescence spectra of the as-grown and annealed binary PFO/SPR and ternary PFO/F8BT/SPR thin films, with various ratios, at the excitation wavelength of 380 nm. In all figures, the PL emission spectra of all pristine films are also plotted, for a better interpretation of the results. The wavelength labels refer to the peaks’ maximums derived by the deconvolution analysis performed in each spectrum [20]. In general, the blue-shifts of the characteristic F8BT and SPR emissions compared to those of the pristine materials can be attributed to the conformational changes in the backbone arrangements of the constituents in the blended polymer films. In the binary PFO/SPR blend thin films, the characteristic PFO emission peaks are preserved, regardless of whether they have been annealed or are in the PFO and SPR blending ratios. On the contrary, the characteristic peak of the SPR emission shows a strong dependence on both the content and on the annealing. More specifically, by increasing the SPR content, the emission intensity in the red region increases more strongly in the as-grown films and, to a lesser extent, in the annealed ones. This can be attributed to the non-effective energy transfer from the PFO matrix to the SPR component due to the formation of the crystalline PFO phase resulting from thermal annealing. More specifically, the crystalline PFO domains act as individual blue emission centers enhancing the ratio of blue to red emission in the normalized PL intensity.

In the ternary films with the addition of the F8BT, the emission peak in the green region, referring to F8BT, dominates the spectrum independently from the F8BT content. This peak is attributed to the efficient energy transfer from the PFO to the F8BT molecules, while the observed blue shifting is attributed to residual PFO emission, as it is justified by Figure 4a. On the other hand, in the as-grown films, the intensity of the main peaks corresponding to PFO decreased dramatically, following the increase in the F8BT content against that of PFO. However, in the annealed ternary films, the PFO emission is preserved in the PFO:F8BT:SPR (99:0.5:0.5) blend, whilst it is reduced at the other two ratios. This indicates that there exists efficient energy transfer from amorphous PFO to the crystalline PFO phase that acts as an efficient blue dopant [45,46]. Furthermore, the broadened PL spectra with additional emission in the long-wavelength region confirm the possibility of dual FRET occurrence in ternary blend thin films from PFO to F8BT and SPR. Thus, the crystalline PFO phase and the F8BT and SPR moieties serve as blue, green, and red fluorescent dopants in the amorphous PFO matrix, making the enhancement of the white emission of the ternary blends possible.

To gain insights into the energy transfer mechanisms, PL spectra were recorded in variable excitations. Figure 6a–d shows the sequential recorded PL spectra, by using excitation wavelengths *λ*^exc^ from 330 to 390 nm with a step of 10 nm, of the as-grown and annealed blends, Binary#1 and Ternary#1. This wavelength range of excitation was selected because it corresponds to the maximum absorption of PFO. It is obvious that in all films, the PL intensity increases with the increase in *λ*^exc^ from 330 to 390 nm as it approaches the energy band gap of the PFO.

### 3.4. Electroluminescence of the Binary and Ternary OLEDs

Figure 7 shows the EL spectra of all devices. The spectra show a broad visible emission from 400 to 800 nm. Devices based on the as-grown binary PFO/SPR blends displayed three emission peaks. The first two peaks at 438 and 518 nm are associated with blue emission from PFO, and the third featureless double peak at 610 and 641 nm is associated with the red emission from SPR. In the case of the annealed binary blends, the enhancement of the PFO emission at 433, 457, and 490 nm, and the remarkable reduction in the SPR emission intensity at ~630 nm for the PFO:SPR (99:1) blend are notable, which shows that the annealing of the films and the subsequent crystallization of the PFO component limits the energy transfer mechanism. With the addition of F8BT in the ternary blends, the dominant emission is located at ~520 nm, which can be attributed to PFO and F8BT combined emissions. For the OLED devices of the as-grown blends of ratios PFO:F8BT:SPR (97:2.5:0.5) and (98:1.5:0.5), an efficient energy transfer from PFO to F8BT is obtained. However, for the (99:0.5:0.5) blend, with the lower F8BT content, the PFO emission is enhanced. By comparing the relative normalized emission intensities in the blue and green range of the annealed blends’ OLED devices (429, 453, 523, and 570), we can conclude that a more balanced energy transfer takes place, and a control in the blue emission intensity can be achieved. This fact may be related to the charge trapping mechanism, that also takes place in EL emission, except for the energy transfer mechanism. This suggests that the carriers being trapped by the crystallized phase of PFO simultaneously occurs in addition to the dual energy transfer mechanism from PFO to F8BT and SPR.

### 3.5. Chromaticity and Electrical Properties

Figure 8 depicts the chromaticity diagram with the Commission Internationale de L’ Eclairage (CIE) coordinates, derived from (a) PL and (b) EL measurements, and the corresponding numerical data are listed in Table 3. Generally, the CIE coordinate visualizes the entire range of colors that can be obtained by mixing the three primary colors (red (R), green (G), and blue (B)) by varying the wavelength and emission intensity. According to the CIE diagram, it is obvious that the PL emission of the ternary blends approaches closer to the white region. Moreover, the thermally treated binary blends show a clear tendency to blue emission. On the contrary, the as-grown ternary blends emit in green, while white emission is achieved after their thermal treatment. Clearly, the tendency of the EL CIE coordinates of the studied blends is similar to that of PL. Annealing causes a blue shift in the emission of the blends. In binary blends, their characteristic yellow-orange emission shifts significantly to the near blue range upon their annealing. Finally, for the case of the ternary blends, the initial green emission of the as-grown ones approaches the desirable white emission upon their thermal treatment. The CIE coordinates of the ternary blends with the higher PFO contents, namely (98:1.5:0.5) and (99:0.5:0.5), are closer to the ideal ones (0.33, 0.33), revealing their potential as efficient white light-emitting materials. In the diagram of Figure 7b, the respective photos of the produced OLED devices have been added, using the binary and ternary as-grown and annealed blends, during their operation.

Concerning the electrical characteristics, the turn-on voltages (measured for a 10 cd/m^2^ luminance) of the OLED devices fabricated using ternary blends are generally lower compared to those of binary ones (see Table 3). This decrease in the current density values may be related to an increase in the exciton confinement and recombination efficiency in the emissive layer, which is important for improving device performance. Furthermore, the Color Rendering Index (CRI) for the majority of the produced devices is high enough, and mainly, for the most promising devices, is satisfactorily high for solution-processed white OLEDs, 70 and above. Finally, the devices exhibit high luminance, especially the ternary blends, although, in general, a reduction in the luminance of the annealed blends is observed compared to the as-grown counterparts. Annealing can probably lead to a phase separation due to the rearrangement of polymer chains [47,48] as well as to an increase in the films’ roughness affecting the device performance by their effect on the Förster energy transfer mechanisms, the charge recombination probability, and the charge injection [49]. Indeed, the luminance values show a declining trend by increasing the PFO content. Nevertheless, a more thorough evaluation of the performance of the materials requires a more systematic study which concerns samples of the same thickness, but this was out of the scope of this work.

## 4. Conclusions

Binary and ternary polymer blends were studied in terms of their photophysical properties and used as photoactive materials for the fabrication of white OLEDs. The binary blends were produced by mixing red SPR into the host blue PFO, whereas the ternary blends were produced by mixing SPR and green F8BT into the host PFO. Different ratios of the constituents, as well as the effect of post growth thermal treatment of the blends, were examined.

It was found that the optical properties and, specifically, the dielectric function, the fundamental band gap, and the optical absorptions of the blends are mainly controlled by the dominant PFO constituent. The correlation of the PFO PL emission spectrum with the absorption spectra of F8BT and SPR demonstrated the potential FRET mechanism from PFO to SPR in the binary blends and the possibility of dual FRET occurrence from PFO to F8BT and SPR in the ternary blends. On the contrary, a negligible overlap between F8BT PL emission and SPR absorption was obtained. The PL emission of both as-grown and annealed binary blends strongly depends on the SPR ratio in the red spectral range. A dominant F8BT PL emission with subsequent elimination of the PFO PL emission for the as-grown ternary blends was obtained. The annealing of the ternary blends enhances the PFO PL emission, which increases by increasing the PFO ratio. From the EL characterization, it was found that FRET is the dominant mechanism in the as-grown blends leading to a characteristic red-orange emission in binary blends and green emission in ternary blends. The PFO emission is preserved in the annealed blends; however, in the binaries, the emission shifts to the near blue region, while, in the ternaries, the emission significantly approaches the white region. The effect of thermal treatment leads to the formation of a crystalline PFO phase in the amorphous PFO matrix, exhibiting an extended conjugation length, as was revealed by the red shift of the absorption. As a result, the crystalline PFO phase acts as an efficient blue dopant for enhancing color purity towards the achievement of white light emission. Furthermore, the electrical characteristics of the produced OLED devices, such as low turn-on voltages (2–3 V) and their operational performances with luminance above 3000 cd/m^2^ and a CRI of 70 and above, demonstrate the use of ternary blends as very promising candidates for white OLEDs.

## Figures and Tables

**Figure 1 nanomaterials-12-04099-f001:**
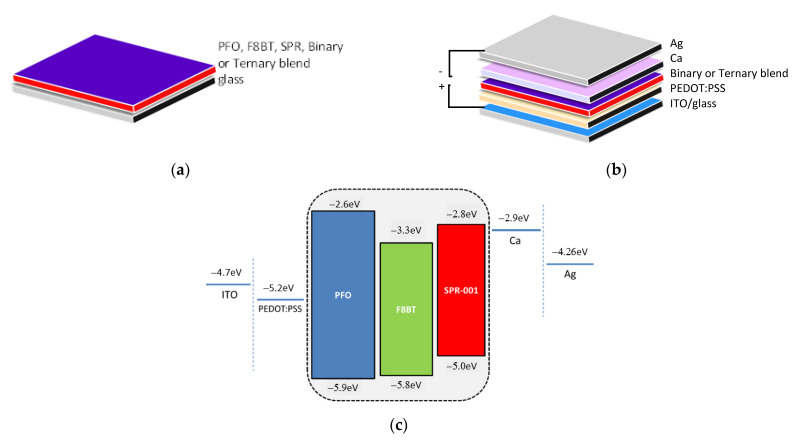
The structure of (**a**) the monolithic blend films, (**b**) the OLED devices, and (**c**) the energy levels of PFO, F8BT, and SPR, which form the blend films, and of the sequential layers of the OLED devices.

**Figure 2 nanomaterials-12-04099-f002:**
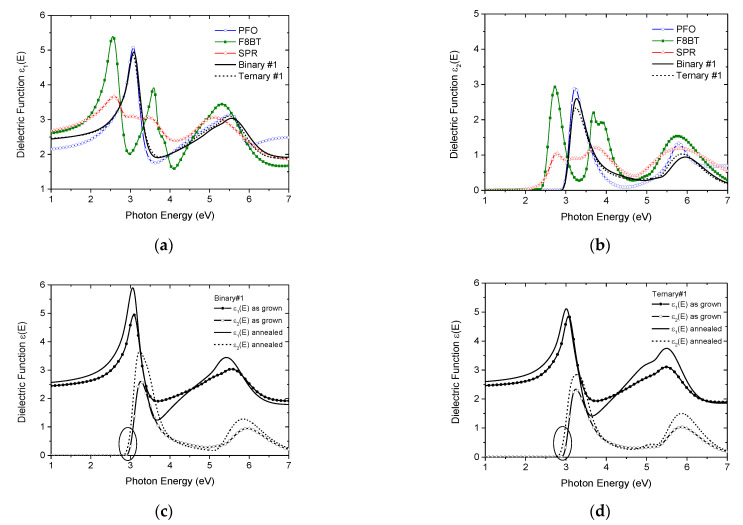
(**a**) The real and (**b**) imaginary part of the dielectric function versus photon energy of the pristine polymers and the two representative investigated blends, Binary#1 and Ternary#1. The dielectric function of the as-grown and annealed (**c**) Binary#1 and (**d**) Ternary#1.

**Figure 3 nanomaterials-12-04099-f003:**
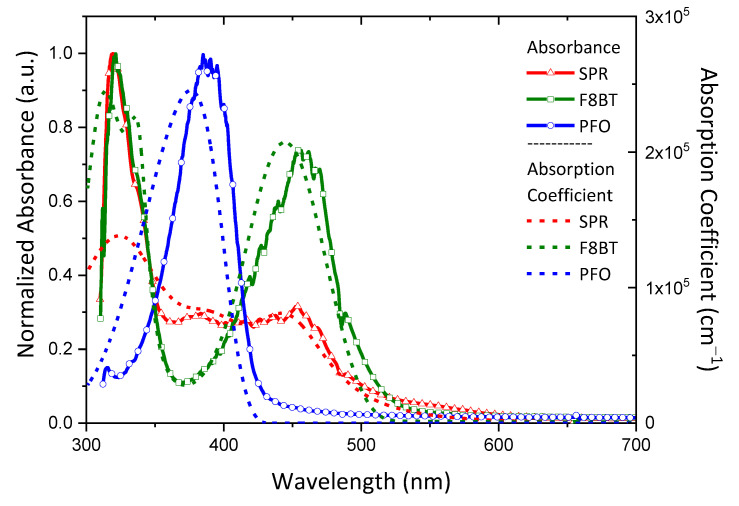
Comparison of the absorption spectra obtained by UV–vis AS (left axis, solid-symbol lines) and the absorption coefficients determined using SE (right axis, dashed lines) of pristine SPR, F8BT, and PFO.

**Figure 4 nanomaterials-12-04099-f004:**
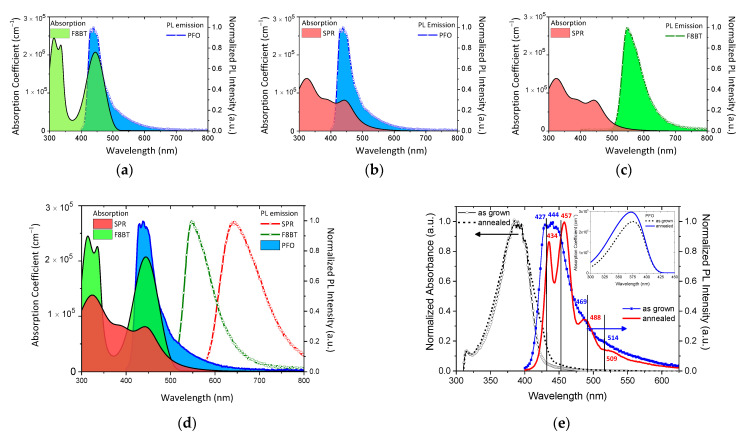
The spectrum overlapped between (**a**) PFO and F8ΒΤ, (**b**) PFO and SPR, and (**c**) F8BT and SPR. (**d**) The overall correlation between the absorption coefficients of the SPR and F8BT components with the PL emissions of PFO, F8BT, and SPR. (**e**) The correlation of Absorbance and PL emission of the as-grown and annealed PFO films and the respective absorption coefficients in the inset.

**Figure 5 nanomaterials-12-04099-f005:**
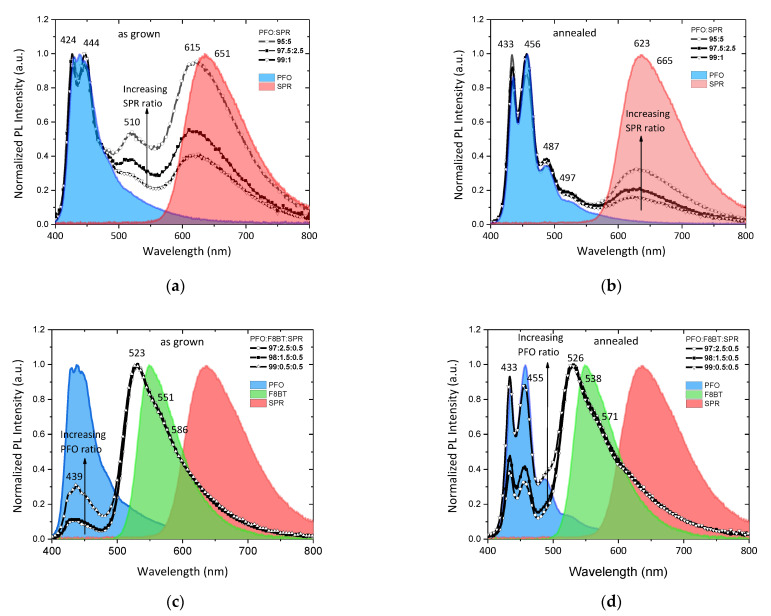
Normalized PL spectra of the (**a**) as-grown, (**b**) thermally treated PFO/SPR binary blends and (**c**) as-grown, (**d**) thermally treated PFO/F8BT/SPR ternary blends. PL emission spectra of all pristine films are also plotted for comparison.

**Figure 6 nanomaterials-12-04099-f006:**
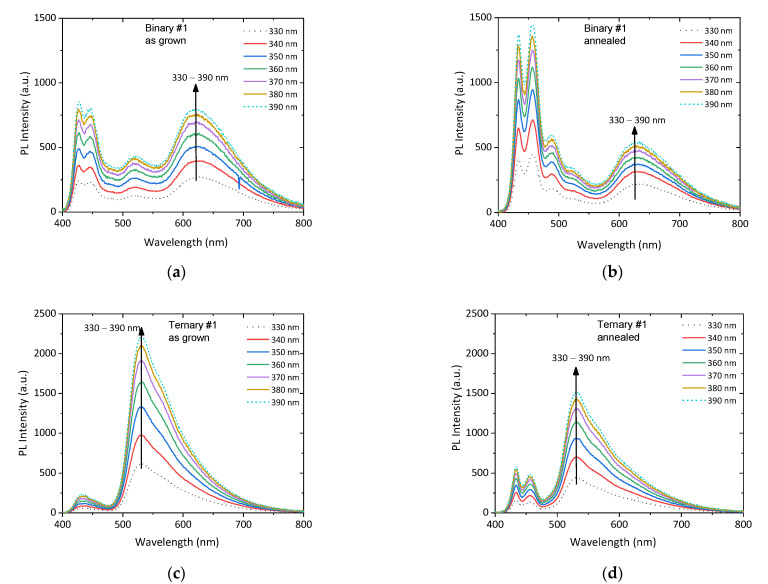
The evolution of PL emission of the Binary#1 (**a**) as-grown and (**b**) annealed and of the Ternary#1 (**c**) as-grown and (**d**) annealed films, with the excitation wavelength *λ*^exc^.

**Figure 7 nanomaterials-12-04099-f007:**
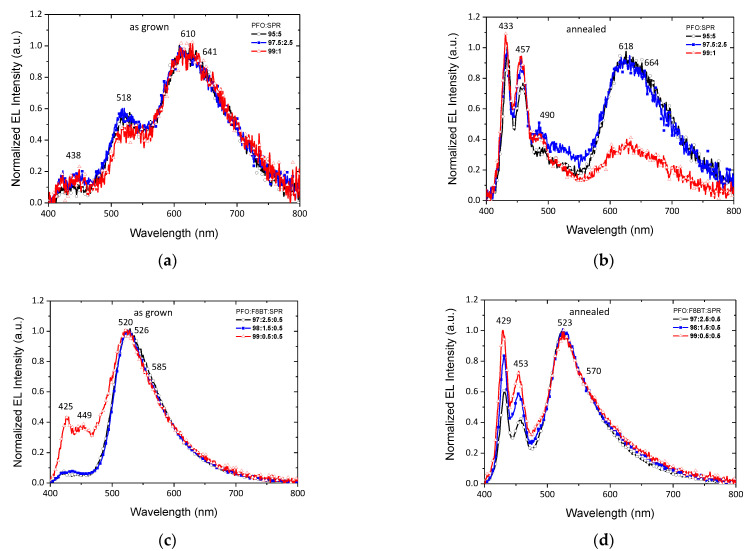
Normalized EL spectra of the (**a**) as-grown, (**b**) thermally treated PFO/SPR binary blends and (**c**) as-grown, (**d**) thermally treated PFO/F8BT/SPR ternary blends.

**Figure 8 nanomaterials-12-04099-f008:**
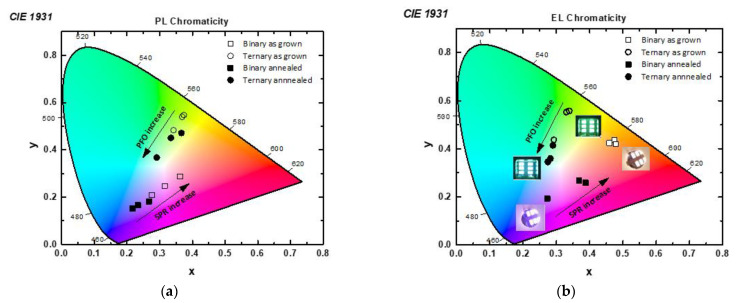
CIE diagrams of (**a**) PL and (**b**) EL emissions of the studied binary and ternary blend films.

**Table 1 nanomaterials-12-04099-t001:** Blend weight ratios and code-names.

Blend Type	Blend Code					
	Binary#1	Binary#2	Binary#3	Ternary#1	Ternary#2	Ternary#3
PFO:SPR	95:5	97.5:2.5	99:1			
PFO:F8BT:SPR				97:2.5:0.5	98:1.5:0.5	99:0.5:0.5

**Table 2 nanomaterials-12-04099-t002:** SE results for the as-grown and thermally treated pristine films and representative binary and ternary blends.

Material/Blend Type		Thickness d (nm)	$E_{g}\;$(eV)	$E_{g}^{Tauc}\;$(eV)	$\Delta E=E_{g}^{Tauc}-E_{g}$ (eV)
SPR	as-grown	73.6	1.53	2.54	1.01
	annealed	72.2	1.56	2.54	0.98
F8BT	as-grown	62.7	2.36	2.55	0.19
	annealed	61.7	2.36	2.55	0.19
PFO	as-grown	46.9	2.87	3.06	0.19
	annealed	48.0	2.88	3.07	0.19
Binary#1	as-grown	48.3	2.88	3.07	0.19
	annealed	56.5	2.79	3.04	0.25
Ternary#1	as-grown	46.3	2.88	3.05	0.17
	annealed	38.5	2.76	2.97	0.21

**Table 3 nanomaterials-12-04099-t003:** The electrical–operational characteristics of fabricated OLED devices.

Blend Type/ Treatment		Turn-on Voltage (V) at 10 cd/m^2^	V at Max Luminance (V)	Max Luminance (cd/m^2^)	CRI	x	y
Binary#1	as-grown	5.5	12	1033	94	0.474	0.436
	annealed	6.0	12	830	57	0.388	0.259
Binary#2	as-grown	3.5	12	1038	94	0.460	0.424
	annealed	6.5	12	721	64	0.369	0.269
Binary#3	as-grown	6.0	12	561	95	0.479	0.419
	annealed	6.5	12	401	50	0.273	0.194
Ternary#1	as-grown	3.0	13	7474	44	0.339	0.557
	annealed	4.0	11	7849	62	0.290	0.415
Ternary#2	as-grown	3.5	14	9838	49	0.345	0.560
	annealed	3.0	12	3614	72	0.275	0.346
Ternary#3	as-grown	3.5	14	3321	74	0.250	0.300
	annealed	2.0	14	3184	70	0.282	0.360

## Data Availability

Data presented in this article are available on request from the corresponding author.

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
