# Peer review of "Influence of Dopant Concentration and Annealing on Binary and Ternary Polymer Blends for Active Materials in OLEDs"

_nanomaterials, 2022, doi:10.3390/nano12224099_

Round 1
Reviewer 1 Report
This manuscript reports the influence of dopant concentration and annealing on OLEDs. The authors fabricated an OLED device with a light-emitting layer composed of a polymer binary and ternary blend and they reported the luminescence characteristics of these films, and they also reported the change in characteristics before and after annealing of these devices. The investigation and conclusion described in this study are sound and promising for WOLED lighting devices development. However, I have some issues with the manuscript as written that must be addressed before it is suitable for publication in Nanomaterials. I recommend a publication after major revision with additional data and a better structuring of the findings.
1. The WOLED references in the “Introduction” part are too old. Of course, the references that reporting the basic theory of WOLED could be old, but authors should refer to the latest papers that reflect newer trends.
2. The thickness of the emission layer shown in Table 2 showed that binary films were increased after annealing, but that for ternary films were decreased after annealing. However, it is well known that the normal heat treatment process reduces the thickness of the film due to the decrease in the free volume of the polymer. The author should explain why the thickness of the binary film was increased after annealing. What is the error range of Ellipsometry? It is known that AFM is the most accurate way to accurately measure a film of this thickness. Since the film thickness changes calculated using spectroscopic ellipsometry were different from the usual known thickness change trend before and after annealing, the authors should compare the exact film thickness change using AFM.
3. According to page 8 line 280 ~282, the authors stated that because of “limitation of energy transfer from the amorphous PFO matrix to the SPR component”, “increasing the SPR content, the emission intensity in the red region increases more strongly in the as grown films and to a lesser extent in the annealed ones.”. Isn't the author's explanation that amorphous PFOs do not effectively transfer energy to SPR? However, as the amount of SPR increases in Fig. 5(a) (i.e. as grown, same as amorphous PFO), the red emission intensity increases significantly, while the amount of SPR increases in Fig. 5(b) (i.e. annealed, same as crystalline PFO), the red emission intensity does not increase significantly. This means amorphous PFO can transfer energy effectively compare to that of crystalline PFO. The authors also wrote in the conclusion of the paper that amorphous PFO causes energy transfer more effectively, so this expression is incorrect and needs correction.
4. According to Fig 4(c), it look like there is a spectral overlap between F8BT and SPR in the range of 500 ~ 550 nm, there is no the cascade energy transfer from F8BT to SPR of the ternary film as shown in Fig. 5(d). The authors should prove that the energy transfer between green and red does not occur through the blend experiment of these two materials.
5. In order to study device characteristics using polymer blend, their morphological change and device performance change should be explained in relation to each other. The authors take AFM images of the polymer blend before and after annealing, and these should be explained in connection with the change in device properties.
Reviewer 2 Report
In this work, white organic LED were fabricated based on binary and ternary blends of materials using spin coating method. Various optical characteristics of the samples were studied using different characterization techniques and the device performance was evaluated. Generally, the manuscript is clear and well-written. The results are well analyzed and are in a good manner. Overall the topic will be of interest to the readers of the journal. Some minor revision to the manuscript is necessary before it is accepted for publication in Nanoelectronics.
1- In the context, there are misspelled words, lack of punctuation, and academic style writing. Therefore, I suggest that the English language needs revision. For instance, in abstract section line 17 "thin films with various ratios were successfully prepared with various ratio"
2- Authors are advised to include some quantitative key findings in the abstract rather than be descriptive.
3- Authors should clarify how they ended up with this particular annealing temperature (120oC)? Does this a random selection or based on a scientific reason? What will be the outcomes if slightly higher or lower temperature was used?
4- In page 6 line 209, it is known that Tauc relation is (αhν)2= A(hν - Eg )x. The value of (x) in the equation can be changed according to the type of electronic transition. Therefore, how can the author confirm that their samples have direct allowed electronic transition and not another type of electronic transition?
5- In page 6 line 218 authors stated that “Furthermore, a systematic decrease in the band gap values, either in the ?g or ??????, is observed between the as grown and thermally treated blends.” However, by looking at table 2, those values are almost constant for as grown and annealed samples. This should be clarified and corrected.
Round 2
Reviewer 1 Report
The revised paper satisfies all deficient points.